# Variational saliency maps for explaining model's behavior

## Abstract

Saliency maps have been widely used to explain the behavior of an image classifier. We introduce a new interpretability method which considers a saliency map as a random variable and aims to calculate the posterior distribution over the saliency map. The likelihood function is designed to measure the distance between the classifier's predictive probability of an image and that of locally perturbed image. For the prior distribution, we make attributions of adjacent pixels have a positive correlation. We use a variational approximation, and show that the approximate posterior is effective in explaining the classifier's behavior. It also has benefits of providing uncertainty over the explanation, giving auxiliary information to experts on how much the explanation is trustworthy.

## 1 Introduction

Since the advent of deep learning brought significant improvement in general machine learning tasks (Krizhevsky et al., 2012), explaining deep networks have become an important issue (Ribeiro et al. (2016)). Problems inherent in training a deep neural network, such as fairness (Arrieta et al., 2020) or the model classifying based on unintended features (Ribeiro et al., 2016), can be mitigated when the model is finely explained. Therefore, the models that have gained users' trust through explanation are preferred in practical applications.

Saliency maps, also called attribution maps or relevance maps, have been widely used for interpretability methods in classification tasks, typically in an image domain (Simonyan et al., 2013). A saliency map represents the importance of each feature of given data that influences the model's decision. There have been several approaches for obtaining the saliency map, which are back-propagation based methods (Ancona et al., 2017; Bach et al., 2015; Lundberg & Lee, 2017; Montavon et al., 2017; Selvaraju et al., 2017; Shrikumar et al., 2017; Simonyan et al., 2013; Smilkov et al., 2017; Srinivas & Fleuret, 2019; Sundararajan et al., 2017) and perturbation based methods (Chang et al., 2019; Chen et al., 2018; Dabkowski & Gal, 2017; Fong et al., 2019; Fong & Vedaldi, 2017; Schulz et al., 2020; Zeiler & Fergus, 2014; Zintgraf et al., 2017). Regardless of the approaches, the common implicit assumption shared by most of the previous interpretability methods is that *a saliency map exists in a deterministic manner when a model and an input data are given*: one attribution map is provided to explain the model's decision for each data point.

Instead of the implicit assumption, we propose a stochastic approach called Variational Saliency maps (VarSal) where it is assumed that the interpretation has inherent randomness. The intuition stems from the stochastic effect that makes interpretation methods more explainable. For instance, FIDO (Chang et al., 2019) expands the search space of the mask by drawing it from Bernoulli distribution. This approach prevents the mask to be searched in the local space when it is directly optimized (Fong & Vedaldi, 2017). The example informs us that the stochastic property draws better interpretation.

We define the posterior distribution as the probability of the saliency map when the training data and the classifier are given. To make the posterior behave as the distribution of explanation, it is essential to carefully design the likelihood function and the prior distribution. We follow the idea of perturbation based methods to form the likelihood where the input that only contains features which correspond to high attribution in a saliency map is likely to describe the classifier's behavior. For modeling the prior, we propose a new covariance matrix of Gaussian distribution that implies the property of having a positive correlation among attributions of adjacent pixels. As this property

mimics total variation (TV) regularization, we name the prior as soft-TV Gaussian prior. After modeling the likelihood and the prior, the Variational Bayesian method (Hoffman et al., 2013; Kingma & Welling, 2013) is used since the posterior is intractable.

After the optimization, unlike most of perturbation based methods, VarSal produces a real-time saliency map since only a single forward pass is required for generating it. Also, the VarSal method provides high quality in the visual inspection where sophisticated borderlines exist with object-oriented attention. We compare VarSal with baseline methods on the perturbation benchmark test to show the effectiveness of our approach. At the end, we examine the benefit of employing a posterior distribution, which is uncertainty over the explanation.

## 2    RELATED WORK

In this section, we take a look at perturbation based interpretability methods. Fong & Vedaldi (2017) optimize the cost function with respect to the mask which indicates the most important features in an image for the classifier's prediction. This approach is further developed by Fong et al. (2019) where they introduce a new method for making a perturbed image which helps to reduce hyper-parameters and produces better qualitative results. Both methods should optimize the mask every time they receive input, which is computationally expensive. Dabkowski & Gal (2017) relax the problem of time complexity by using a trained network of which the output is a saliency mask. However, all three methods have a limitation for producing importance ranking among features of a given image since their objective is to produce a binary mask.

PDA (Zintgraf et al., 2017) produces a saliency map from a different perspective. It computes the importance of each pixel by regarding it as an unobserved pixel and marginalizes it out to get the predictive probability output of the classifier. The same idea is used in FIDO (Chang et al., 2019) to generate a perturbed image that is regarded as a sample from training data distribution. It optimizes the parameters of a Bernoulli dropout distribution for making a saliency mask. It helps exploring the search space of binary mask rather than being limited to local search since the mask is sampled from the distribution for each training iteration. Our method is similar to FIDO in that VarSal also explores the search space by sampling the saliency map from the encoder in the training phase.

There is an information theoretic approach for explaining the classifier's prediction. Schulz et al. (2020) adopt an information bottleneck for restricting the flow of information in an intermediate layer by adding noise. They find the importance of each feature by calculating the information flow. Chen et al. (2018) also adopt mutual information concept and optimize its variational bound for training a network that maps an input image to a saliency map. VarSal is similar in that we also train the encoder network by optimizing the evidence lower bound (ELBO). However, our method differs in that we regard the saliency map as a random variable and aim to calculate the posterior over the saliency map.

## 3    VARIATIONAL SALIENCY MAPS

In this section, we introduce details of the VarSal method which provides stochastic saliency maps. Let us define a pre-trained classifier that we aim to interpret as $M : \mathbb{R}^{c \times h \times w} \rightarrow \mathcal{Y}$ where $\boldsymbol{x} \in \mathbb{R}^{c \times h \times w}$ is an input with $c$, $h$, and $w$ to be channel, height, and width of the input image, respectively, and $\mathcal{Y} = \{1, 2, \ldots, K\}$ is a set of classes. The classifier $M$ provides categorical probability $P_M(\cdot) = \hat{\boldsymbol{y}} \in \triangle^{K-1}$ where $\triangle^{K-1}$ is a $K-1$ simplex. Since the purpose of a saliency map $\boldsymbol{s} \in \mathbb{R}^{h \times w}$ is to describe the behavior of the classifier's prediction, our goal is to calculate the posterior distribution of the saliency map, $p(\boldsymbol{s}|\boldsymbol{x},\hat{\boldsymbol{y}})$ (solid lines in Figure 1). By Bayes' rule, the posterior is stated as:

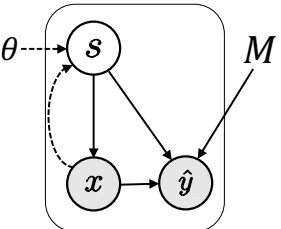

Figure 1: Graphical model.

$$p(\boldsymbol{s}|\boldsymbol{x},\hat{\boldsymbol{y}}) = p(\hat{\boldsymbol{y}}|\boldsymbol{x},\boldsymbol{s})\ p(\boldsymbol{s}|\boldsymbol{x})\ /\ \mathrm{Z}\,, \qquad (1)$$

where Z is the marginal likelihood. To calculate the posterior, we should model two terms: the likelihood $p(\hat{\boldsymbol{y}}|\boldsymbol{x},\boldsymbol{s})$ and the prior $p(\boldsymbol{s}|\boldsymbol{x})$.

## 3.1 MODELING LIKELIHOOD

The likelihood should be well-designed to make the posterior over the saliency map explain the behavior of the classifier. We focus on the property that the importance of each feature in an image is determined by observing the response of the classifier's output when the feature is perturbed (Zeiler & Fergus, 2014; Fong & Vedaldi, 2017; Zintgraf et al., 2017). More specifically, important features are enough to correctly classify the input as target class with high confidence. This concept is first introduced by Dabkowski & Gal (2017), and called smallest sufficient region (SSR). The difference between SSR and our approach is that we do not consider the *smallest* sufficient region, but rather *rank* the features (therefore, $s \in \mathbb{R}^{h \times w}$, not $s \in \{0, 1\}^{h \times w}$). Moreover, we consider not the *target class* but the *categorical probability* to interpret the model itself. The likelihood is designed such that the aforementioned properties satisfy

$$- \log p(\hat{\boldsymbol{y}}|\boldsymbol{x}, \boldsymbol{s}, k) = D_{\mathrm{KL}}[\, P_M(\boldsymbol{x}) \parallel P_M(\boldsymbol{x} \odot \tau^{(k)}(\boldsymbol{s})) \,] + \mathrm{const}, \tag{2}$$

$$p(\hat{\boldsymbol{y}}|\boldsymbol{x}, \boldsymbol{s}) = \mathbb{E}_{p(k)}[\, p(\hat{\boldsymbol{y}}|\boldsymbol{x}, \boldsymbol{s}, k) \,], \tag{3}$$

where $D_{\mathrm{KL}}$ is a Kullback-Leibler (KL) divergence, $\tau^{(k)}$ is a top-$k$ operation, and $\odot$ is a perturb operation that makes local perturbation of input $\boldsymbol{x}$. The top-$k$ operation applied to the saliency map, $\tau^{(k)}(\boldsymbol{s}) \in \{0, 1\}^{h \times w}$, acts as a mask where $[\tau^{(k)}(\boldsymbol{s})]_{i,j} = 1$ when $\boldsymbol{s}_{i,j}$ corresponds to the biggest $k$ attributions in $\boldsymbol{s}$. This way, the top-$k$ operation makes the conditional likelihood in equation 2 to consider only the selected features in the input. By varying $k$, the amount of selected features is controlled, and we set $p(k)$ as uniform distribution. To make the local perturbation of input $\boldsymbol{x}$ using perturb operation $\odot$, we follow the method proposed by Fong & Vedaldi (2017), $\boldsymbol{x} \odot \boldsymbol{m} = \boldsymbol{x} \circ \boldsymbol{m} + \tilde{\boldsymbol{x}} \circ (1 - \boldsymbol{m})$, where $\tilde{\boldsymbol{x}}$ is a baseline input, and $\circ$ is a pointwise multiplication. We bring three baseline settings in our experiment: blurred baseline[1], noise baseline[2], and mean baseline[3].

The equation 2 states that the categorical probability $\hat{\boldsymbol{y}}$ is more likely when the distance between the classifier's predictive probability of input $\boldsymbol{x}$ and that of perturbed input is close for given $k$. This makes sense since better saliency map that explains the classifier's behavior would approximate the model's prediction closer with the selected features of top-$k$ attributions. Also, we do not consider the ground-truth class or the top-1 predicted class, but rather whole classes with predictive probability in order to examine the classifier's behavior itself. To consider various values of $k$, we also take the expectation in equation 3.

## 3.2 SOFT-TV GAUSSIAN PRIOR

The easiest way to model the prior distribution $p(\boldsymbol{s}|\boldsymbol{x})$ is to consider it as independent of $\boldsymbol{x}$, $p(\boldsymbol{s}|\boldsymbol{x}) = p(\boldsymbol{s})$, and design it as standard Gaussian distribution $\mathcal{N}(\mathrm{vec}(\boldsymbol{s}); \boldsymbol{0}, \boldsymbol{I})$ where $\mathrm{vec}(\boldsymbol{s}) \in \mathbb{R}^{hw}$ is a vectorized version of $\boldsymbol{s}$. However, the standard Gaussian prior does not consider the belief over the saliency map that the attribution of neighbor pixels might have correlation (Fong et al., 2019). Therefore, we propose a new prior distribution that expresses the belief. While Dabkowski & Gal (2017); Fong & Vedaldi (2017) proposed total variation (TV) regularization to prevent the saliency mask from being occurred adversarial artifacts, we mimic the TV method in building the prior distribution to provide the positive correlation between neighbor attributions. To be more specific, we design the prior distribution as zero mean Gaussian distribution, $\mathcal{N}(\mathrm{vec}(\boldsymbol{s}); \boldsymbol{0}, \boldsymbol{\Sigma})$, and infuse the TV knowledge to the covarianace matrix by setting $\boldsymbol{\Sigma}_{i,j} > 0$ when pixel $i$ and pixel $j$ are identical or adjacent. We have

$$\boldsymbol{\Sigma}_{i,j} = \begin{cases} 1, & \text{if } i = j \\ \alpha, & \text{if } i - j \in \{-w, -1, 1, w\} \text{ and } j \in \mathrm{Adj}_i \\ \alpha^2, & \text{if } i - j \in \{-w-1, -w+1, w-1, w+1\} \text{ and } j \in \mathrm{Adj}_i \\ 0, & \text{otherwise}, \end{cases} \tag{4}$$

---

[1] We define "blurred baseline" as an input image blurred with Gaussian kernel.

[2] The "noise baseline" is defined as Gaussian noise.

[3] We term "mean baseline" when the baseline is set to be the per channel mean of an original image and added by Gaussian noise.

where $\alpha > 0$ and $\mathrm{Adj}_i$ is the adjacent index set of pixel $i$ (Figure 9 in Appendix B). This way, we grant the TV knowledge to the Gaussian prior, and call it soft-TV Gaussian prior.

## 3.3 Variational inference on saliency maps

Modeling the likelihood as equation 3 and the prior as $\mathcal{N}(\mathrm{vec}(\boldsymbol{s}); \boldsymbol{0}, \boldsymbol{\Sigma})$ makes the posterior of equation 1 intractable. Therefore, we approximate it with the distribution $q_\theta(\boldsymbol{s}|\boldsymbol{x})$ parameterized by $\theta$ (dotted lines in Figure 1) where the objective is to minimize the KL divergence between $q_\theta(\boldsymbol{s}|\boldsymbol{x})$ and $p(\boldsymbol{s}|\boldsymbol{x}, \hat{\boldsymbol{y}})$:

$$
\begin{aligned}
&\operatorname*{argmin}_\theta\ D_{\mathrm{KL}}[\,q_\theta(\boldsymbol{s}|\boldsymbol{x})\ \|\ p(\boldsymbol{s}|\boldsymbol{x},\hat{\boldsymbol{y}})\,]\\
&= \operatorname*{argmin}_\theta\ \mathbb{E}_q[\underbrace{-\log p(\hat{\boldsymbol{y}}|\boldsymbol{x},\boldsymbol{s})}_{(*)}] + \underbrace{D_{\mathrm{KL}}[\,q_\theta(\boldsymbol{s}|\boldsymbol{x})\ \|\ p(\boldsymbol{s}|\boldsymbol{x})\,]}_{(**)}\,.
\end{aligned}
\tag{5}
$$

We apply a mean-field approximation with univariate Gaussian for each factorized term of approximate posterior, $q_\theta(\boldsymbol{s}|\boldsymbol{x}) = \mathcal{N}(\,\mathrm{vec}(\boldsymbol{s});\ \boldsymbol{\mu}_\theta(\boldsymbol{x}), \mathrm{diag}(\boldsymbol{\nu}_\theta(\boldsymbol{x}))\,)$, where $\boldsymbol{\mu}_\theta(\cdot) \in \mathbb{R}^{hw}$ is the mean of the distribution and $\mathrm{diag}(\boldsymbol{\nu}_\theta(\cdot)) \in \mathbb{R}^{hw \times hw}$ is the diagonal covariance matrix with the main diagonal to be $\boldsymbol{\nu}_\theta(\cdot) \in \mathbb{R}^{hw}$. $\boldsymbol{\mu}_\theta$ and $\boldsymbol{\nu}_\theta$ are collectively called encoder network parameterized by $\theta$. The encoder network is optimized with the training dataset used for training the classifier $M$, and the reparameterization trick (Kingma & Welling, 2013) is applied during optimization. The schematic description is shown in Appendix A.

There are two problems in optimizing the equation 5: non-differentiable top-$k$ operation in equation $(*)$ and computationally expensiveness in equation $(**)$. In case of $(*)$, the top-$k$ operation $\tau^{(k)}$ is non-differentiable where the gradient cannot flow backward. To overcome this issue, we approximate it with a differentiable SOFT operator proposed by Xie et al. (2020). This allows flowing the gradient from the classifier $M$ to the encoder parameters $\theta$. Note that the classifier $M$ is a pre-trained classifier that we aim to interpret, and thus should be fixed.

Since the size of covariance $\boldsymbol{\Sigma}$ is large, which is $hw \times hw$, it is computationally expensive to calculate $(**)$ when it is naively used (in case of Imagenet dataset (Russakovsky et al., 2015), the size of $\boldsymbol{\Sigma}$ is $224^4$ !). We solve the problem by decomposing $\boldsymbol{\Sigma}$ with a Kronecker product:

$$
\boldsymbol{\Sigma} = \boldsymbol{\kappa}_h \otimes \boldsymbol{\kappa}_w\,,
\tag{6}
$$

where $\otimes$ is the Kronecker product, and $\boldsymbol{\kappa}_h \in \mathbb{R}^{h \times h}$ and $\boldsymbol{\kappa}_w \in \mathbb{R}^{w \times w}$ are tridiagonal matrices with 1 for main diagonal and $\alpha$ for first diagonal below and above the main diagonal (Figure 9 in Appendix B). After all, $(**)$ can be analytically derived as:

$$
\begin{aligned}
D_{\mathrm{KL}}[\,q_\theta(\boldsymbol{s}|\boldsymbol{x})\ \|\ p(\boldsymbol{s}|\boldsymbol{x})\,] =\ &\mathrm{diag}\left(\boldsymbol{\kappa}_w^{-1}\right)^T \cdot \mathrm{rsh}\left(\boldsymbol{\nu}_\theta\right) \cdot \mathrm{diag}\left(\boldsymbol{\kappa}_h^{-1}\right) - \mathrm{sum}\left(\log \boldsymbol{\nu}_\theta\right)\\
&+ \mathrm{sum}\left(\mathrm{rsh}\left(\boldsymbol{\mu}_\theta\right) \odot \left(\boldsymbol{\kappa}_w^{-1} \cdot \mathrm{rsh}\left(\boldsymbol{\mu}_\theta\right) \cdot \left(\boldsymbol{\kappa}_h^{-1}\right)^T\right)\right) + \mathrm{const}\,,
\end{aligned}
\tag{7}
$$

where $\mathrm{sum}(\cdot)$ is the summation of elements. For a vector $\boldsymbol{b} \in \mathbb{R}^{hw}$, we denote $\mathrm{rsh}(\boldsymbol{b}) \in \mathbb{R}^{w \times h}$ as reshaping the vector $\boldsymbol{b}$ to the matrix where $[\mathrm{rsh}(\boldsymbol{b})]_{i,j} = \boldsymbol{b}_{i+wj}$. Also, for a square matrix $\boldsymbol{B}$, $\mathrm{diag}(\boldsymbol{B})$ is the vector where the $i$th entry is $\boldsymbol{B}_{i,i}$. The derivation of equation 7 is provided in Appendix D. The equation 7 shows that the computation is easily done in general deep learning frameworks such as Pytorch (Paszke et al., 2017).

## 4 Experiment

### 4.1 Implementation detail

There are two terms in the objective function equation 5: the reconstruction term (*) and the regularization term (**). The components in the regularization term have high dimension, which is $hw$. This is usually too large that the regularization term becomes dominant in the loss function.

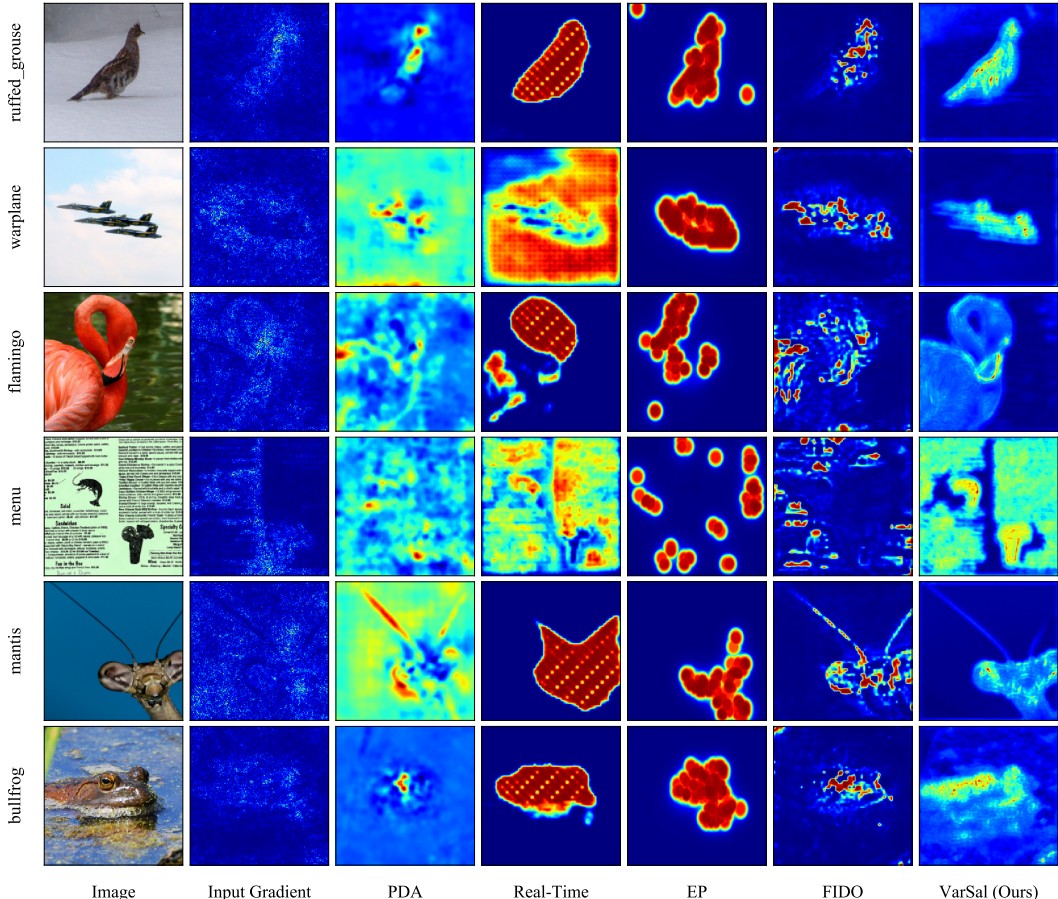

Figure 2: Qualitative results. Compared with previous methods, VarSal visually captures the most sophisticated borderlines and object-oriented saliency map.

We avoid this phenomenon by introducing a hyper-parameter $\beta$ to the regularization term (Higgins et al., 2017). We set the default value of $\beta$ to be $1/(10hw)$.

We test our method on the ImageNet dataset (Russakovsky et al., 2015). We use a pre-trained VGG16 (Simonyan & Zisserman, 2014) for the classifier $M$, and 16 convolutional layers for the encoder $\boldsymbol{\mu}_\theta$ and $\boldsymbol{\nu}_\theta$. As for the variable $k$ used in the top-$k$ operator, we sample $k$ from the uniform distribution, $k \sim \mathcal{U}(hw/10, 9hw/10)$, for each training iteration. The baseline input $\tilde{x}$ is also randomly selected among three baselines (blurred, noise, and mean baseline) for each training iteration. As for $\alpha$ in the soft-TV Gaussian prior, we set $\alpha = 0.4$. Finally, if not mentioned, the mean of approximate posterior is used when performing qualitative and quantitative experiments. More details are given in Appendix C.

## 4.2 QUALITATIVE RESULTS

We perform visual inspection by comparing VarSal with previous interpretability methods. For fair comparison, with the data that are correctly classified, previous interpretability methods are performed based on the ground-truth target, while VarSal explains the predictive probability. In Figure 2, the first heatmap (Simonyan et al., 2013) are generated by the gradients of input. They visually highlights the object, but there exists sparsity. PDA (Zintgraf et al., 2017) usually provides unnecessary highlights since it only considers spatially local parts in the optimization process. Real-time saliency (Dabkowski & Gal, 2017) and EP (Fong et al., 2019) show a boolean mask for the saliency map with smoothed borderlines. This is because it optimizes the mask with the size smaller than the input, followed by performing upsampling for the final saliency mask. FIDO (Chang et al.,

2019) provides the shape of the object to some extent, but it still has sparsity. Compared to the previous approaches, VarSal does not contain upsampling process, but rather directly provides the saliency map with the same size of input image. As the last heatmap shows, the VarSal highlights the object with more sophisticated borderlines. More results are provided in Appendix E.

To investigate the importance of modeling the prior distribution, we perform visual inspection between the VarSal method trained with standard Gaussian prior and that of soft-TV Gaussian prior. As Figure 3 shows, VarSal trained with standard Gaussian prior provides saliency map with high frequency noise inside the object boundary. This is because the standard Gaussian prior does not constrain the adjacent attributions to have correlation. On the other side, VarSal optimized using the soft-TV Gaussian prior shows smaller variation of attribution between adjacent pixels. Moreover, the noise on the background has been reduced when the soft-TV prior is used for modeling the prior distribution.

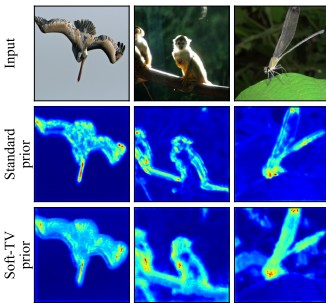

Figure 3: Prior selection.

### 4.3 SANITY CHECK

The prerequisite for becoming an interpretability method is to pass the sanity check (Adebayo et al., 2018). This is to identify whether the interpretability method provides a saliency map dependent of a classifier or a data, and is tested by randomizing the classifier's parameters. The difference between saliency maps obtained by the original classifier $M$ and the parameter-randomized classifier is measured by structural similarity index (SSIM) and Spearman rank correlation. It is known that Guided Back-

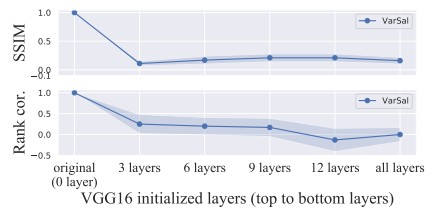

Figure 4: Sanity check.

Prop (Springenberg et al., 2014) and Guided GradCAM (Selvaraju et al., 2017) do not pass the sanity check since the SSIM and the Spearman metric is close to 1. Instead, Figure 4 shows that our method has lower values for each metric and different randomization of the classifier, indicating that VarSal has passed the sanity check.

### 4.4 QUANTITATIVE RESULTS

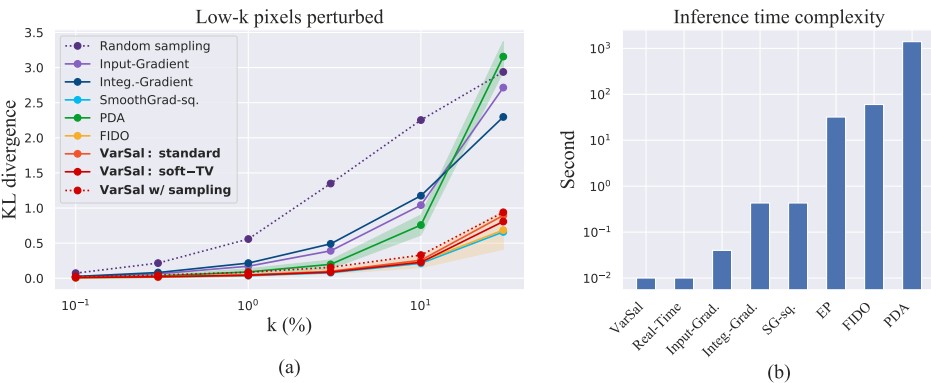

Figure 5: (a) Quantitative results for pixel perturbation benchmark The results verify the usefulness of our method. (b) Time complexity of each interpretability methods. SG-sq. refers to SmoothGrad-squared.

Determining the state-of-the-art interpretability method is challenging since there is no evaluation benchmark that exactly reflects the method's performance (Hooker et al., 2019). Commonly used benchmarks can split the superiority and inferiority of each method to some extent, but cannot exactly rank them based on the quantitative evaluation indicator. We use pixel perturbation benchmark to verify the usefulness of our method.

For the pixel perturbation metric, image pixels are erased that correspond to the largest $k\%$ saliency values (Ancona et al., 2017; Samek et al., 2016) or the smallest $k\%$ saliency values (Srinivas & Fleuret, 2019), and observe the response to the change of classifier's output. In our experiment, we erase pixels with the latter procedure as the former is more prone to create unnecessary artifacts that lead to misunderstanding of the reason for the score drop (Srinivas & Fleuret, 2019). We change pixel values of the input image to zero that correspond to the least $k\%$ values in the saliency map, and observe the KL divergence between the classifier's predictive probability of original input and that of perturbed input. The interpretability method is thought to be better when the distance is smaller.

It takes expensive computational time for PDA and FIDO method, about 25 minute and 1 minute per one image (Figure 5(b)). Therefore, for evaluating PDA and FIDO, we randomly sample 100 data in the validation dataset to perform the top-k perturbation benchmark, with repeating the process 5 times. Other methods use entire validation dataset for the evaluation. For fair comparison, saliency map is generated from the predicted class for each interprebility methods except VarSal. Also, perturbing randomly drawn k pixels is suggested as a control experiment. We omit drawing error range for this control experiment since the error (standard deviation) is less than 1e-2. The results are shown in Figure 5(a). It is observed that Input-Gradient (Simonyan et al., 2013), Integrated-Gradient (Sundararajan et al., 2017), and PDA gets close to the control setting as $k$ gets larger. For others such as SmoothGrad-squared (Smilkov et al., 2017; Hooker et al., 2019), FIDO, and VarSal, they have similar values throughout the change of $k$ values. Moreover, VarSal performs better when using the soft-TV Gaussian prior than using the standard Gaussian prior.

An interesting point is that the saliency map obtained by sampling from the approximate posterior instead of the mean values results in low-quality (red dotted line in Figure 5(a)). We speculate this is because the sampling method produces artifacts in a perturbed image that cause score drop (Kurakin et al., 2016). As shown in Figure 6, even though both the sampled saliency map and the mean saliency map captures objects with high values, the sampled one has high-frequency noise. When the image pixels corresponding to low $k\%$ of the noisy saliency map are perturbed, the perturbed image has the artifact of sharp color contrast between adjacent pixels that might cause score degradation.

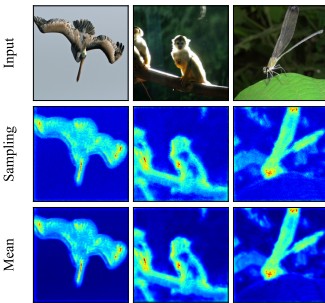

Figure 6: Sampled saliency map.

### 4.5 UNCERTAINTY OVER EXPLANATION

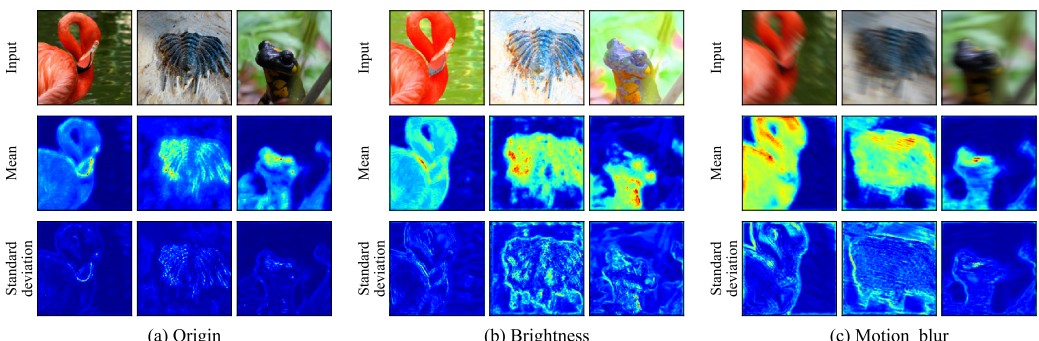

(a) Origin      (b) Brightness      (c) Motion_blur

Figure 7: Uncertainty over the explanation. The posterior distribution of saliency map gives two summaries, which are the explanation (mean) and the uncertainty over the explanation (standard deviation). They are compared qualitatively on the original dataset and two different shifted dataset.

One advantage of having posterior distribution over the saliency map is that it gives us the uncertainty of the explanation. This is done by summarizing the posterior with its covariance matrix. Since the VarSal approximates the posterior with factorized Gaussian, we can observe the variance

of each attribution where the examples are shown in Figure 7(a). While the explanation (second row) has higher attribution at the object, the uncertainty over the explanation (third row) presents at the borderline of the object.

We also qualitatively compare the posterior results on the shifted samples. Recently Hendrycks & Dietterich (2018) have established Imagenet-C dataset that is created by visually corrupting the data in the Imagenet dataset. The first row in Figure 7(b) and (c) respectively shows the example of images corrupted by "Brightness" and "Motion blur" with level 5 of severity. While the explanation of corrupted images is similar to that of original images in that they all capture the object to some extent, the uncertainty of the explanation shows different appearance where the heatmaps of standard deviation are noisy in the corrupted images.

## 5    CONCLUSION AND FUTURE WORK

In this paper, we presented a new perspective on a saliency map where it is assumed to be a random variable. After designing the likelihood function and the prior distribution that makes the posterior distribution over the saliency map explain the behavior of the classifier's prediction, the approximate posterior is optimized by maximizing ELBO. The experimental results were performed with the mean of the approximate posterior, and showed that our method has visually sharp borderlines with object-oriented saliency map. For quantitative results, the pixel perturbation benchmark is used to prove the effectiveness of our method. We verified that using the proposed soft-TV Gaussian distribution rather than the standard Gaussian distribution for modeling the prior has better performance in both qualitative and quantitative comparison. It was also shown that the proposed VarSal method has a strong advantage over other methods in terms of inference computation complexity. Finally, we showed that our method provides not only the explanation but also the uncertainty over the explanation.

There remain future works for producing better quality of the posterior distribution over a saliency map. In modeling the likelihood, the problem of data distribution shift could be mitigated by generating a perturbed image that is expected to be sampled from training data distribution (Chang et al., 2019). It would also be interesting to consider axiomatic prior knowledge (Sundararajan et al., 2017; Srinivas & Fleuret, 2019) in modeling the prior distribution and the likelihood. Finally, the uncertainty consideration over explaination is believed to be critical since it can tells us how much the explanation given by an interpretability method can be trusted, which needs to be studied further.

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

APPENDIX

## A    SCHEMATIC DESCRIPTION

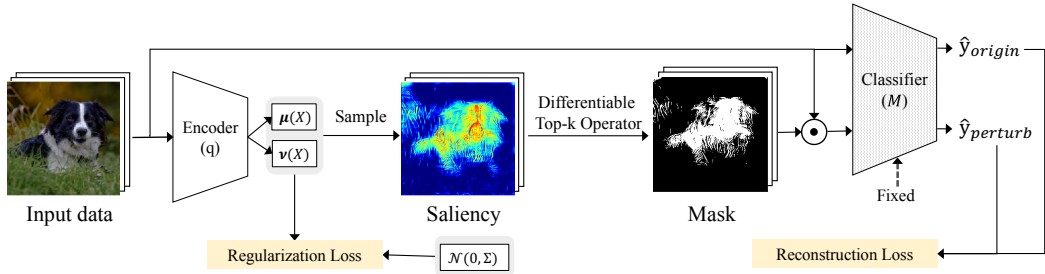

Figure 8: Schematic description.

An input image is fed into the encoder network that gives the mean and the variance of Guassian distribution (which is the approximate posterior). A saliency map is sampled from it, followed by passing a differentiable top-$k$ operator to provide a binary mask. With this mask, the input image is perturbed. The perturbed image is passes a classifier $M$ that gives categorical probability $\hat{\boldsymbol{y}}_{perturb}$. The loss function is composed of two terms: the reconstruction term between $\hat{\boldsymbol{y}}_{perturb}$ and the categorical probability obtained from the original image $\hat{\boldsymbol{y}}_{origin}$, and the regularization term between the approximate posterior $q(\boldsymbol{s}|\boldsymbol{x}) = \mathcal{N}(\boldsymbol{s}; \boldsymbol{\mu}(\boldsymbol{x}), \mathrm{diag}(\boldsymbol{\nu}(\boldsymbol{x})))$ and the prior distribution $\mathcal{N}(\boldsymbol{s}; \boldsymbol{0}, \boldsymbol{\Sigma})$. As the classifier $M$ is the model that we aim to interpret, it is fixed so as the parameter not to be updated while training the encoder network.

## B  SOFT-TV GAUSSIAN PRIOR

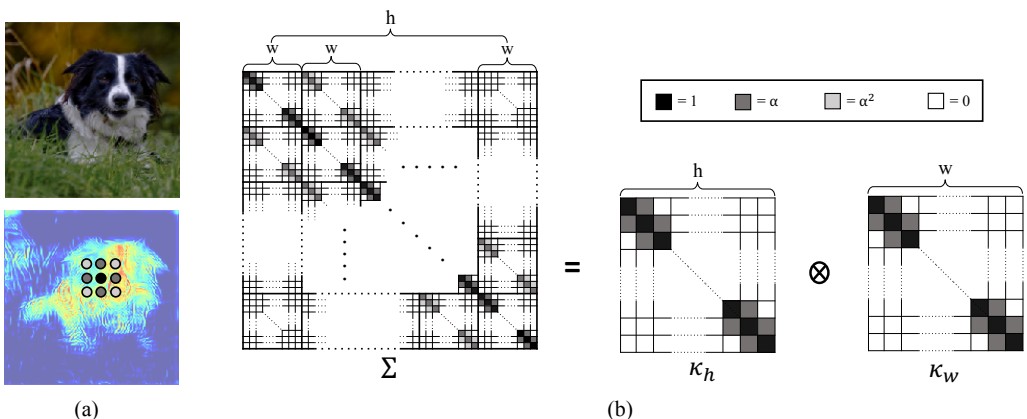

Figure 9: Soft-TV Gaussian prior.

We consider the prior knowledge that adjacent pixels in a saliency map have positive correlation (Figure 9(a)). After modeling the prior as Gaussian distribution $\mathcal{N}(\boldsymbol{s}; \boldsymbol{0}, \boldsymbol{\Sigma})$, the covariance matrix $\boldsymbol{\Sigma}$ is designed to infuse this prior knowledge into the prior distribution (Figure 9(b)). The covariance matrix is then decomposed by Kronecker product to better calculate the KL divergence of the regularization loss.

## C  IMPLEMENTATION DETAILS

**Encoder architecture**   We use 17 convolution layers for the encoder network. To make the spatial size of the encoder's input and output to be same, we do not use a pooling layer. Every convolution layer is comprised of a convolution with kernel size $3 \times 3$, stride 1, and padding 1, followed by batch normalization and a rectified linear unit. The number of output channels for each convolution is as follows: $[64, 64, 64, 64, 32, 32, 32, 32, 32, 16, 16, 16, 16, 16, 16, 2]$. The encoder network provides $\boldsymbol{\mu} \in \mathbb{R}^{h \times w}$ and $\eta \in \mathbb{R}^{h \times w}$ for each channel of the output where $\boldsymbol{\mu}$ is the mean of the Gaussian distribution and $\eta = \log \nu$ with $\nu \in \mathbb{R}^{h \times w}$ the variance of the Gaussian distribution.

**hyper-parameters**   We use Adam (Kingma & Ba, 2014) optimizer with learning rate to be $0.0001$, weight decay to be $0.0005$, and betas to be $(0.9, 0.99)$. We use batch size of $128$ while training the encoder network. We run 10 epochs for the Imagenet dataset, and save the network that has the lowest loss.

## D  PROOF OF REGULARIZATION LOSS EQUATION

Let us first define the notation. $\otimes$ is the Kronecker product, and $\odot$ is the element-wise multiplication. $\mathrm{sum}(\cdot)$ is the summation of elements. For a vector $\boldsymbol{b} \in \mathbb{R}^{hw}$, we denote $\mathrm{rsh}(\boldsymbol{b}) \in \mathbb{R}^{w \times h}$ as reshaping

the vector $\boldsymbol{b}$ to the matrix where the $(i, j)$-th entry is $\boldsymbol{b}[i + wj]$, and $\mathrm{diag}(\boldsymbol{b})$ as the diagonal matrix where the diagonal is $\boldsymbol{b}$. Also, for a square matrix $\boldsymbol{B}$, $\mathrm{diag}(\boldsymbol{B})$ is the vector where the $i$-th entry is $B[i, i]$. For a matrix $\boldsymbol{B}$, $\mathrm{vec}(\boldsymbol{B})$ denotes the vectorization by stacking the columns of the matrix $\boldsymbol{B}$ to a single column vector.

Recall that the approximate posterior is $q(\boldsymbol{s}|\boldsymbol{x}) = \mathcal{N}(\boldsymbol{\mu}, \boldsymbol{\Sigma}_0)$ where $\boldsymbol{\Sigma}_0 = \mathrm{diag}(\boldsymbol{\nu})$ with $\boldsymbol{\mu}, \boldsymbol{\nu} \in \mathbb{R}^{hw}$, and the prior distribution is $p(\boldsymbol{s}|\boldsymbol{x}) = \mathcal{N}(\boldsymbol{0}, \boldsymbol{\Sigma}_1)$ where $\boldsymbol{\Sigma}_1 = \boldsymbol{\kappa}_h \otimes \boldsymbol{\kappa}_w$ with $\boldsymbol{\kappa}_h \in \mathbb{R}^{h \times h}$ and $\boldsymbol{\kappa}_w \in \mathbb{R}^{w \times w}$. KL divergence between two Gaussian distribution is:

$$
\begin{aligned}
D_{\mathrm{KL}}[\, q(\boldsymbol{s}|\boldsymbol{x}) \,\|\, p(\boldsymbol{s}|\boldsymbol{x}) \,] &= D_{\mathrm{KL}}[\, \mathcal{N}(\boldsymbol{\mu}, \boldsymbol{\Sigma}_0) \,\|\, \mathcal{N}(\boldsymbol{0}, \boldsymbol{\Sigma}_1) \,] \\
&= \frac{1}{2}\left( \mathrm{tr}\left( \boldsymbol{\Sigma}_1^{-1} \boldsymbol{\Sigma}_0 \right) + \boldsymbol{\mu}^T \boldsymbol{\Sigma}_1^{-1} \boldsymbol{\mu} - hw + \log \frac{|\boldsymbol{\Sigma}_1|}{|\boldsymbol{\Sigma}_0|} \right).
\end{aligned}
\tag{8}
$$

We compute each term in RHS of equation 8 to make it computationally efficient.

$$
\begin{aligned}
\mathrm{tr}\left( \boldsymbol{\Sigma}_1^{-1} \boldsymbol{\Sigma}_0 \right) &= \mathrm{tr}\left( (\boldsymbol{\kappa}_h \otimes \boldsymbol{\kappa}_w)^{-1} \cdot \mathrm{diag}(\boldsymbol{\nu}) \right) \\
&= \mathrm{tr}\left( (\boldsymbol{\kappa}_h^{-1} \otimes \boldsymbol{\kappa}_w^{-1}) \cdot \mathrm{diag}(\boldsymbol{\nu}) \right) \\
&= \mathrm{diag}\left( \boldsymbol{\kappa}_h^{-1} \otimes \boldsymbol{\kappa}_w^{-1} \right) \odot \mathrm{diag}(\boldsymbol{\nu}) \\
&= \mathrm{diag}\left( \boldsymbol{\kappa}_w^{-1} \right)^T \cdot \mathrm{rsh}(\boldsymbol{\nu}) \cdot \mathrm{diag}\left( \boldsymbol{\kappa}_h^{-1} \right).
\end{aligned}
\tag{9}
$$

$$
\begin{aligned}
\boldsymbol{\mu}^T \boldsymbol{\Sigma}_1^{-1} \boldsymbol{\mu} &= \boldsymbol{\mu}^T \cdot \left( \boldsymbol{\kappa}_h^{-1} \otimes \boldsymbol{\kappa}_w^{-1} \right) \cdot \boldsymbol{\mu} \\
&= \boldsymbol{\mu}^T \cdot \left( \boldsymbol{\kappa}_h^{-1} \otimes \boldsymbol{\kappa}_w^{-1} \right) \cdot \mathrm{vec}\left( \mathrm{rsh}(\boldsymbol{\mu}) \right) \\
&= \boldsymbol{\mu}^T \cdot \mathrm{vec}\left( \boldsymbol{\kappa}_h^{-1} \cdot \mathrm{rsh}(\boldsymbol{\mu}) \cdot \boldsymbol{\kappa}_w^{-1T} \right) \\
&= \mathrm{sum}\left( \mathrm{rsh}(\boldsymbol{\mu}) \odot \left( \boldsymbol{\kappa}_w^{-1} \cdot \mathrm{rsh}(\boldsymbol{\mu}) \cdot \left( \boldsymbol{\kappa}_h^{-1} \right)^T \right) \right).
\end{aligned}
\tag{10}
$$

$$
\begin{aligned}
\log \frac{|\boldsymbol{\Sigma}_1|}{|\boldsymbol{\Sigma}_0|} &= \log |\boldsymbol{\kappa}_h \otimes \boldsymbol{\kappa}_w| - \log |\mathrm{diag}(\boldsymbol{\nu})| \\
&= \log |\boldsymbol{\kappa}_h|^w |\boldsymbol{\kappa}_w|^h - \log \prod_{i=1}^{hw} \nu_i \\
&= w \cdot \log |\boldsymbol{\kappa}_h| + h \cdot \log |\boldsymbol{\kappa}_w| - \log \prod_{i=1}^{hw} \nu_i \\
&= -\mathrm{sum}(\log \nu_i) + \mathrm{const}.
\end{aligned}
\tag{11}
$$

Therefore, the equation 8 is derived as:

$$
\begin{aligned}
D_{\mathrm{KL}}[\, q(\boldsymbol{s}|\boldsymbol{x}) \,\|\, p(\boldsymbol{s}|\boldsymbol{x}) \,] = {}& \mathrm{diag}\left( \boldsymbol{\kappa}_w^{-1} \right)^T \cdot \mathrm{rsh}(\boldsymbol{\nu}) \cdot \mathrm{diag}\left( \boldsymbol{\kappa}_h^{-1} \right) \\
&+ \mathrm{sum}\left( \mathrm{rsh}(\boldsymbol{\mu}) \odot \left( \boldsymbol{\kappa}_w^{-1} \cdot \mathrm{rsh}(\boldsymbol{\mu}) \cdot \left( \boldsymbol{\kappa}_h^{-1} \right)^T \right) \right) \\
&- \mathrm{sum}(\log \nu_i) + \mathrm{const}.
\end{aligned}
\tag{12}
$$

# E    QUALITATIVE RESULTS

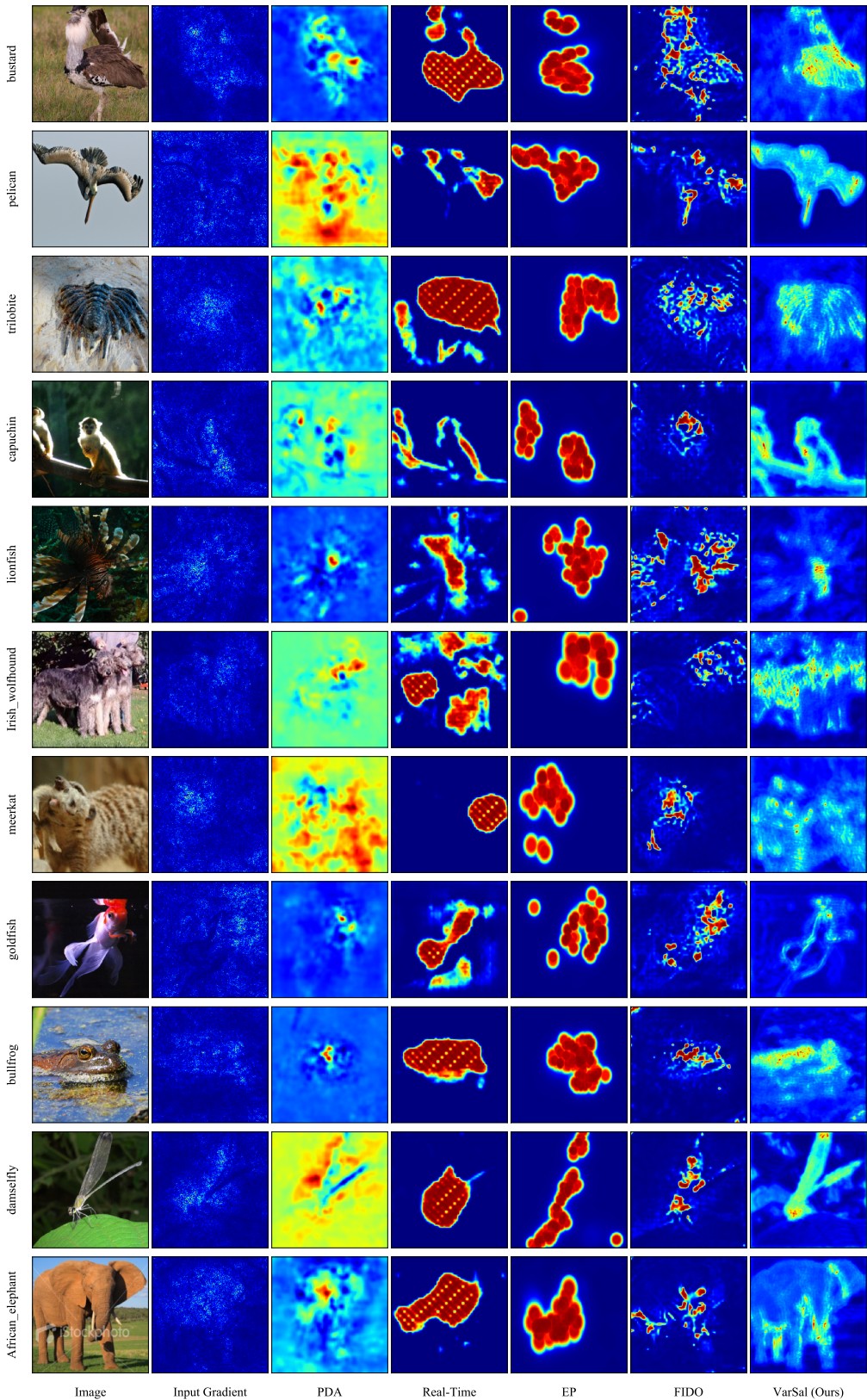

