# OpenReview forum: "Variational saliency maps for explaining model's behavior"
_ICLR.cc/2021/Conference — Reject_

### Official Review · AnonReviewer3 · 2020-10-27
**Interesting idea, not fully evaluated**

**Rating:** 4
**Confidence:** 3

**Review:**

Summary: This paper proposed a method for generating saliency maps for image classifiers that are stochastic (instead of deterministic). The probabilistic model assumes a saliency map random variable that generates the data with a classifier. The inference is done by variational methods. The paper presents several qualitative examples and a comparison to previous work using the pixel perturbation benchmark.

Strengths:
- Saliency map generation is an important problem of interest to the ICLR community.
- Stochastic saliency map is interesting and novel to my knowledge.
- The paper is generally easy to understand.

Concerns:
- Current empirical experiments compare to several saliency generation methods in the pixel perturbation benchmark. The authors should also compare to FullGrad and use the remove-and-retrain framework as in (Srinivas & Fleuret, 2019). This will help us fully understand the performance of this method.
- A positive point is the improved inference time. This is shown in Figure 5b but not discussed in the text, only briefly mentioned in the conclusion. I also think that improved inference time is at the cost of longer training time (the authors can correct me on this).
- A central claim is that stochastic saliency maps improve explanation. The authors have provided some intuition and examples, but I think more evidence is needed.
- The presentation of the paper can be improved: the abstract and the introduction should highlight the contributions and summarize the main results, which the experiment section should fully discuss; the sanity check and implementation details can be cut or moved to supplementary.

---

> ### Author Response · Authors · 2020-11-25
> **Response to Reviewer 3**
>
> We first thank the reviewer 3 for providing constructive feedback.
>
> Q) empirical experiments
>
> A) Since the objective of the proposed method is to understand the behavior of model’s prediction which is predictive probability (not ground-truth target), we cannot test on the remove-and-retain (ROAR) metric. However, we can slightly change our method to test ROAR and compare our method with previous approaches. We set our method suggested in W3 by Reviewer2, which is to change the objective to explain the ground-truth target rather than the predictive probability.
>
> Testing on {10,30,50,70,90}% pixel perturb, we observed that Input-gradient, Integrated-gradient, and SmoothGrad has a few performance degradtion while VarSal (ours), SmoothGrad-Squared, and VarGrad has significant drop. For example, when 90% of pixels are perturbed, VarSal achieved 20% accuracy while the accuracy by Input-Gradient, Integrated-gradient, and SmoothGrad was over 60% (lower is better). In case of SmoothGrad-Squared and VarGrad, it was about 10%. However, we we claim that this does not mean our model is worse than SmoothGrad-squared and VarGrad. To verify this argument, we perform ROAR with the center-squared perturbation method, which refers to the perturbation of the center of an image with the shape of square. For all of the training and validation images, center-square has no meaning for interpretability since it is just a spatially fixed region. However, we got 21% of accuracy when 90% of pixels are perturbed, which is considerably low. Therefore, we claim that the ROAR benchmark can provide superiority of each interpretability method to some extent by observing the accuracy degradation, but cannot compare the performance between methods that significantly drop the accuracy.
>
> ---
>
> Q) Trade-off
>
> A) It is true that our method needs additional time for training the interpretability network. However, in situations where real-time inference is required for interpretability, test time would be much more important. We would discuss the strength of our method on inference time in the final version of the paper.
>
> ---
>
> Q) The presentation of the paper
>
> A) Following the suggestions, we will improve the presentation, highlight the contribution and summarize results in introduction and abstract section, and discuss more about the experiments.

---

### Official Review · AnonReviewer4 · 2020-10-29
**A interpretability method with variational approximation for image classification networks**

**Rating:** 5
**Confidence:** 4

**Review:**

Overview:
This paper proposes a new interpretability method for image classification networks. It considers a saliency map as a random variable and aims to calculate the posterior distribution over the saliency map.  The likelihood function and the prior distribution are then designed to make the posterior distribution over the saliency map explain the behavior of the classifier’s prediction. Quantitative evaluation on the perturbation benchmark as well as qualitative result show the effectiveness of the proposed method over baselines.

Pro:
+ The proposed method is very efficient at the inference stage after the training with extra a encoder.
+ From results on the perturbation benchmark, the proposed method achieves better or comparable performance compared with baseline methods.

Concerns:
- The evaluation is a kind of weak. Since it is hard to evaluate the interpretability method on one single benchmark, why not following existing explanation works e.g. Schulz et al. (2020) to conduct evaluations on multiple tasks with different metrics and network backbone. These tasks could be object localization, image degradation (Schulz et al. 2020), etc.
- For the current evaluation on perturbation benchmark, the proposed method is comparable and sometime worse than some recent perturbation-based methods such as SmoothGrad-squared, FIDO. Although the proposed method is quite efficient at the inference stage, it requires extra training steps and extra encoder with parameters. It is a kind of trade-off compared with other  perturbation-based methods.
- The selected visualization examples have quite simple background. Is it possible to show example with complicated scenes such as samples from PASCAL VOC, MSCOCO, Visual Genome datasets.
- It is not clear whether the proposed method can be generalized to other kinds of data rather than images, such as text data, video data, vision-language tasks etc.

---

> ### Author Response · Authors · 2020-11-25
> **Response to Reviewer 4**
>
> We first thank the reviewer 4 for providing constructive feedback.
>
> Q) Evaluation metric
>
> A) We agree with the statement that it would be more robust if we perform evaluation on different metrics and backbone networks. We would add additional results on ResNet50 and GoogleNet backbone to the final version. Also, we compared interpretability methods on object localization evaluation metric, and showed that ours are worse than Input-Gradient. However, we think that localization metric is not a proper metric since bounding box localization covers lots of background pixels as well as foreground. Actually, if we compare qualitative results between VarSal (ours) and Input-Gradient, we can observe that the saliency map by Input-Gradient has high attribution on the background.
>
> ---
>
> Q) Trade-off
>
> A) It is true that our method needs additional time for training the interpretability network. However, in situations where real-time inference is required for interpretability, test time would be much more important.
>
> ---
>
> Q) Complicated samples
>
> A) We tested several images that contain both the cat and the dog. The network trained by ImageNet is used for the test. Since the objective of our method is to understand the behavior of model’s prediction which is predictive probability (not ground-truth target), the saliency map used to highlight the object of the class that model predicts. For example, if the top-1 prediction is “chihuahua”, the saliency map has high attribution at dog’s face. Also, if the second highest prediction is “tiger_cat”, the saliency map also highlights the cat’s fur.
>
> ---
>
> Q) Generalization to other kinds of data
>
> A) Since our method trains an encoder without fine-tuning a backbone network that we aim to explain, it does not have to be the vision task for performing VarSal. For example, if the objective is to explain the model that is trained by text data, we can train the encoder by perturbing the embedding vector to zero. Therefore, VarSal is model-agnostic and system-agnostic method.

---

### Official Review · AnonReviewer2 · 2020-10-31
**Interesting idea but concerns with methodology and evaluation**

**Rating:** 4
**Confidence:** 3

**Review:**

Summary:

The paper presents a new saliency map interpretability method for the task of image classification. It considers the saliency map as a random variable and computes the posterior distribution over it. The likelihood measures the predictions of the classifier for an image and its perturbed counterpart. The prior encodes positive correlation among adjacent pixels. Variational approximation is used to approximate the posterior.

————————————————————————————————————————————————————————————————————————————————


Strengths:

S1) The paper is very well written and easy to understand.

S2) The paper does a good job of combining the ideas of perturbation based saliency map methods and total variation regularization with variational approximation in proposing their interpretability approach.

S3) The proposed approach is able to generate real-time saliency maps, and, therefore, is computationally cheap.

S4) The paper shows that their proposed interpretability method passes the sanity check from [Adebayo et al. (2018)].

————————————————————————————————————————————————————————————————————————————————


Weaknesses:

W1) My biggest concern with the paper is that the proposed approach entails training another (non-interpretable) network to explain a given pretrained classifier. One of the goals of interpretability is to ensure that the pretrained classifier doesn’t encode any undesired biases learnt from the data. How can we ensure that this newly trained network doesn’t encode any biases of its own, especially when this network is also trained on the same data?


W2) Another key concern is the evaluation of the proposed approach. Details below:

W2a) I am not sure if I understand the importance of qualitative examples correctly. The paper claims that VarSal is able to produce high quality object borderlines as compared to other methods. Is that a desirable property we should expect from interpretability methods? Ideally, we want the saliency maps to highlight regions which the classifier considered the most important. It is not clear to me how we can evaluate that from qualitative examples.

W2b) Why is the case where pixels with the largest k% saliency values are erased more prone to creating unnecessary artifacts than the case where pixels with the smallest k% saliency values are erased? Especially, when compared to other methods, is there a reason to believe that these artifacts might be more common in some methods than others?

W2c) Why is Real-Time [Dabkowski & Gal (2017)] omitted from the pixel perturbation benchmark (Figure 5a)? Given that it is computationally equally cheaper to the proposed approach, comparison with Real-Time seems to be the most important one.

W2d) The proposed approach has many similarities to [Chen et al. (2018)]. How does it compare to the proposed approach / what are the advantages of the proposed approach over [Chen et al. (2018)]?

W2e) The paper in its current form doesn’t provide an explicit take-away message. Given a number of saliency map based interpretability methods available at this point, why should a researcher choose the proposed approach over other works? I think it would be a good idea for the authors to discuss the advantages as well as the disadvantages explicitly, highlighting applications/tasks/conditions where their approach might be better suited than others as well as cases where it might be better to avoid the proposed approach.


W3) A very clear difference between the proposed approach and previous works is that the proposed approach explains the predicted probability distribution while the previous works focus on the ground-truth target. The difference is clearly an advantage of the proposed approach. However, it is important to disassociate the contribution of this difference from the rest of the approach. Is it possible for the authors to create an ablation of their approach keeping this aspect same as the previous works and observe how that version compares (to previous works as well as the proposed approach) qualitatively and quantitatively?

W4) Saliency maps highlight the parts of the input image that the classifier finds most important while making a prediction. In that sense, isn’t it more reasonable that for a given classifier and an image, the saliency map is deterministic? What does it mean intuitively for this saliency map to have randomness? It would be great if the authors can discuss why it makes more sense to treat it as a random variable instead of a deterministic quantity.

W5) Can the authors please describe the construction of the graphical model in Figure 1? Shouldn’t there be a solid line from ‘M’ to ’s’?

W6) The paper says that [Fong & Vedaldi (2017)], [Fong et al. (2019)] and [Dabkowski & Gal (2017)] -- “all three methods have a limitation for producing importance ranking among features of a given image since their objective is to produce a binary mask”. What is the limitation here? And, how is the proposed method overcoming that limitation?

W7) The take-away message from the uncertainity over explanation subsection in unclear. What is the significance of the uncertainity saliency maps? And, what should one learn by generating these maps for the corrupted datasets?

W8) How is the hyperparameter alpha in the soft-TV Gaussian prior selected?


——————————————————————————————————————————————————————————————
——————————————————————————————————————————————————————————————

Update after rebuttal: I thank the authors for their responses to all my questions. They satisfactorily answer some of my concerns. However, I still have two major concerns: 1) the faithfulness of the proposed approach, and 2) I see the potential contribution of uncertainty saliency maps but without an application/evaluation, their significance is unclear. I disagree that uncertainty/confidence generated from the same mechanism that generated the explanation is more trustworthy than the explanation itself. Hence, I cannot recommend the paper for acceptance.

---

> ### Author Response · Authors · 2020-11-25
> **Response to Reviewer 2**
>
> A3) For the ablation study, we experimented by replacing predictive probability \hat{y} with ground-truth target y in defining the posterior distribution in equation (1). In case of quantitative results, we tested on lowest-k% perturbation benchmark. The fraction of logit variation is measured, and lower value is considered better. The result is as follows: FullGrad < VarSal (ours) < Smoothgrad-Squared < SmoothGrad < PDA < input-Gradient < Integrated-Gradient for 0.1 <= k <= 10. For k=10% the value was FullGrad: 0.06, VarSal: 0.08, Smoothgrad-Squared: 0.09, SmoothGrad: 0.21, PDA: 0.22, Input-Gradient: 0.24, Integrated-Gradient: 0.26. The result shows that our method still has competitive performance.
>
> ---
>
> A5) We considered the model ‘M’ has a saliency map (or latent variable) that describes the reason for providing ‘\hat{y}’ when input ‘x’ is given. In that case, there should be a solid line from ‘M’ to ’s’. Thanks for pointing this out. We will describe the construction of the graphical model in the final version.
>
> ---
>
> A6) What we tried to say is that the domain of saliency maps for three existing approaches and that of ours is different. While the former generates binary mask, the latter provides attribution of real value. We think the word ‘limitation’ in the sentence of the main paper is not proper. We will fix it in the final version.
>
> ---
>
> A8) We performed grid search between 0 and 0.5 for finding the best hyper-parameter alpha. The reason we constrain the alpha to be less then 0.5 is to make a covariance in equation (4) to be positive semi-definite matrix.

---

> ### Author Response · Authors · 2020-11-25
> **Response to Reviewer 2**
>
> We first thank the reviewer 2 for providing constructive feedback.
>
> A1) Undesired bias is a common phenomenon among perturbation-based methods. For example, in case of MP and Real-Time, artificial (adversarial) interpretation comes out when optimized without a regularization term, which can also be considered as undesired bias. While the above methods try to solve this problem by the total variation loss, we try to solve it through the soft-TV Gaussian prior and the sampling from the posterior distribution for each training iteration. Although the encoder itself is composed of a neural network and thus non-interpretable, we emphasize that the undesired bias when explaining the classifier is mitigated through the stochasticity of the encoder.
>
> ---
>
> A2a) The intent we try to convey through the qualitative results is that the VarSal method combines the expression styles that other methods have. The saliency map by VarSal highlights the object area like EP or Real-Time does while drawing the borderline like FIDO. Since we do not know what the true interpretation is, we cannot argue that these properties are desirable as the reviewer mentioned. But at least we can compare the qualitative results between other methods. We do not argue that VarSal is qualitatively better than others, but rather that VarSal has characteristic of capturing objects.
>
> ---
>
> A2b) In case of erasing top k% pixels, larger “drop” of logit (or probability) is thought of as a better saliency method. The score can significantly drop for the adversarial artifacts without erasing the (true) important pixels. Instead if we consider erasing smallest k% drop, better interpretability is believed to have smaller “increase” of logit value. Even though it still causes artifacts, the test is to measure how less the logit changes even if they have large artifacts.
>
> To back up this empirically, we tested on ImageNet dataset by removing top 10% pixels and obtained fraction of logit variation. SmoothGrad: 0.53, SmoothGrad-squared: 0.45, Input-Gradient: 0.44, FullGrad: 0.35. Here, larger is better, and it is rather inexplicable (counter-intuitive) to see that SmoothGrad is better than SmoothGrad-squared, and Input-Gradient is better than FullGrad (while this may be true, researchers in this field may disagree). Instead for smallest 10% pixels perturbed, the results are more plausible. Here, smaller is better, and the results are shown as: (FullGrad: 0.08) < (Smoothgrad-squared: 0.1) < (SmoothGrad: 0.2) < (Input-Gradient: 0.23).
>
> ---
>
> A2c) In case of Real-Time, it does not provide ranking among features, but rather gives a binary mask. Therefore, we cannot evaluate it with top-k (and low-k) perturbation benchmark.
>
> ---
>
> A2d) Even though VarSal (ours) and L2X [Chen et al. (2018)] have similar objective function, VarSal tries to calculate the posterior while L2X tries to maximize the mutual information. Another difference is whether to fix the number k which chooses the number of pixels to perturb. While L2X fixes the number k for training the explainer, VarSal do not fix it in order to rank features. Moreover, L2X introduces trainable neural network that approximates the model that aims to explain while we use the model itself.
>
> ---
>
> A2e, A4, A7) While lots of interpretability methods have been proposed, It is quite hard for practitioners to decide which one to use since there are no true label for the saliency map and no clear evaluation metrics to assess them. If there is such a problem, the issue would be alleviated if there is an interpretability method that also provides “confidence level to its explanation”. The uncertainty over given explanation might help practitioners decide whether to believe the explanation, thus believe the model’s decision. This is why we think stochastic saliency maps are needed.

---

### Decision · Program_Chairs · 2021-01-07
**Final Decision**

**Decision:**

Reject

**Comment:**

Overall the reviewers had various positive things to say about the paper, including that it was well written and easy to understand, topical, that the method was sensible, novel and interesting and that the computational efficiency (i.e. real time) was appealing.  However, all the reviewers thought it wasn't quite ready for acceptance, mainly citing concerns with the empirical evaluation.  It seems they had trouble interpreting the empirical results and placing the work with respect to other relevant methods.

It seems in the author response, the authors did much to add to the experiments, but ultimately the reviewers were not comfortable with acceptance.  Taking the reviewers' feedback into account and adding the desired empirical evaluation would make this a much stronger submission to a subsequent conference.